# Post-Pandemic Patient Safety Culture: A Case from a Large Metropolitan Hospital Group in Taiwan

**DOI:** 10.3390/ijerph18094537

**Published:** 2021-04-24

**Authors:** Hsing Yu Chen, Luo Lu, Yi Ming Ko, Jui Wen Chueh, Shu Ya Hsiao, Pa Chun Wang, Cary L. Cooper

**Affiliations:** 1Department of Obstetrics and Gynecology, Taipei City Hospital and Musoon Women’s and Children’s Clinic, Taipei 10491, Taiwan; hychen2005@gmail.com; 2Department of Business Administration, National Taiwan University, Taipei 10617, Taiwan; ming80706@gmail.com; 3Medical Quality Management Center, Taipei City Hospital, Taipei 10341, Taiwan; A3330@tpech.gov.tw (J.W.C.); A3561@tpech.gov.tw (S.Y.H.); 4Joint Commission of Taiwan, Taipei 22069, Taiwan; drtony@seed.net.tw; 5Alliance Manchester Business School, University of Manchester, Manchester M15 6PB, UK

**Keywords:** patient safety culture, safety attitudes questionnaire (SAQ), exhaustion, COVID-19

## Abstract

Patient safety is the core goal of medical institutions. The present study focuses on the patient safety culture and staff well-being admit the COVID-19 pandemic. In a large metropolitan hospital group, 337 employees who had participated in the quality improvement interventions completed an anonymous questionnaire of patient safety culture and personal well-being. The multiple regression analyses indicated that managerial role, seniority, female gender and direct contact with a patient were significantly related to the positive attitude on overall or certain dimensions of safety culture. Multivariate analysis also found that dimensions of teamwork climate, safety climate, job satisfaction and stress recognition as patient safety culture predicted staff exhaustion. Finally, comparing with the available institutional historic data in 2018, the COVID group scored higher on the working condition dimension of patient safety culture, but lower on the stress recognition dimension. The COVID group also scored higher on exhaustion. In the post-pandemic era, there seems to be an improvement on certain aspect of the patient safety culture among hospital staff, and the improvement is more prevalent for managers. However, exhaustion is also a poignant problem for all employees. These findings can inform hospital decision-makers in planning and implementing future improvements of patient safety culture and promoting employee well-being and resilience. Our findings also reveal directions for future research.

## 1. Introduction

Medical institutions are places to take care of patients’ health and treat diseases. Ensuring patient safety should be one of the core goals. Research has shown that while relevant rules, policies, procedures and training often existed, they were not sufficient to change corresponding behavior. Therefore, organizational climate approaches have been employed to highlight social aspects of the work environment, making certain characteristics more salient to employees, thus cueing a change toward desired behaviors. A successful example is the induction of information security climate in medical facilities [1]. Similarly, having a good patient safety culture can reduce medical errors, improve patient prognosis and reduce the length of hospital stay and related medical expenses [2]. Furthermore, it can also reduce the propensity of medical disputes, ameliorate psychological pressure and work stress on employees, thus preventing turnover by mitigating burnout, especially emotional exhaustion [3]. In sum, research has shown that a good patient safety culture can contribute to patient outcomes, effectiveness of medical institutions and the well-being of hospital staff [2].

The definition of safety culture is “the product of individual and group values, attitudes, perceptions, competencies and patterns of behavior that determine the commitment to, and the style and proficiency of, an organization’s health and safety management [4]”. With increasing awareness of the importance of hospital-wide patient safety culture, tools have been developed to assess staff safety attitudes and inform initiatives to improve patient safety culture in health-providing organizations. Among these tools, the University of Texas Safety Attitudes Questionnaire (SAQ) [5] is the most widely used internationally and in Taiwan, where we conducted the present study [6]. The underlying theoretical view of the SAQ recognizes that many medical accidents and adverse incidents are not simply any individual’s errors. The root cause is the latent failures embedded in the organization and system [7,8]. Thus, a distinct advantage of the SAQ is its systematic assessment of a multifold of factors related to medical risk and errors, including organizational, work environment, team and staff factors [8]. In Taiwan, the Joint Commission of Taiwan (JCT), an independent accreditation body for health care institutions, acts as the national leader in patient safety improvements and began its nationwide annual survey using SAQ to chart patient safety culture in all ranges of hospitals, and monitor the long-term change trajectories in the country [6]. Thus, another advantage for using the SAQ is the availability of database for comparison to position our COVID-19 cohort against its own historic institutional performance. Despite the considerable interest and wide use of the SAQ, a recent review has raised concerns for the quality of measurement tools, level of analysis, and outcome measures [2]. In the present study, we thus expand the scope to include staff burnout (exhaustion) as a critical indicator of individual well-being (the reverse of strain), which in aggregate contributes to staff turnover at the organizational level [3,9].

Burnout is defined as a combination of emotional exhaustion, cognitive weariness and physical fatigue and is clearly related to work stressors and strains [10]. Emotional exhaustion (or briefly exhaustion), characterized as feelings of energy depletion or exhaustion, is the most important and widely studied dimension in the burnout literature [11]. Burnout remains a persistent issue affecting healthcare staff worldwide [3,12]. Despite the urge to expand the SAQ research to include more outcome measures [13], very few studies have examined the relation between staff exhaustion and patient safety culture. One study found that staff exhaustion affected the patient safety culture, and it was mainly related to factors at the team level of the work, rather than the individual level [14]. Evidence is also very limited about the direct impact of staff burnout on patient outcomes. One recent study reported that higher odds of patient mortality, failure to rescue and prolonged length of stay were found in US hospitals that had, on average, higher nurse burnout scores. The same study also noted that good work environments attenuated the relationships between nurse burnout and mortality, failure to rescue and length of stay. The SAQ encompasses dimensions assessing the work environment in terms of job satisfaction, perception of management and work conditions. It is poignant that improving the work environment could help hospitals to simultaneously improve staff well-being and patient outcomes.

During the COVID-19 pandemic, the burden on and challenges to the medical system and personnel has been unprecedented. The uncertainties about the diagnosis and treatment of this unknown emerging disease, unfamiliarity with new tasks due to redeployment and changes in models of care delivery, increasing workload and restrictions on and off work can all cause stress. At the same time, the medical staff must protect themselves from infection and avoid infecting their family members. Research has shown that working in such challenging conditions can hinder the ability of hospital staff to deliver safe and effective care, amplify exhaustion, contributing to poor patient safety [15]. Although Taiwan has a very low COVID-19 death toll (11 deaths of today, 21 April 2021), heath workers are at greater risk of being exposed to death at close quarters, and may suffer from post-traumatic stress disorder [16]. Scholars have warned that working in the unprecedented pandemic situations may cause some health care workers to experience moral injuries or mental health problems, such as depression, post-traumatic stress disorder and even suicidal ideation [16]. A latest systematic review of studies on the impact of COVID-19 on mental health indeed found that health care workers suffered from increased depression/depressive symptoms, anxiety, psychological distress and poor sleep quality [17]. A recent comparative study in Italy and Spain, the two countries badly hit by the COVID-19, found no cross-country differences of grief during the pandemic [18]. It is thus imperative to monitor the state of well-being of health care workers, irrespective of the country they are working in, in order to support the frontline soldiers in this long haul fight against the pandemic.

At the beginning of the COVID-19 outbreak, the Taiwanese government swiftly decided to close the border (31 December 2019), execute a strict 14-day quarantine (21 March 2020), make mask wearing mandatory in public and introduce other mass prevention measures. Owing to hard lessons learned from the previous 2003 SARS epidemic, the decisive activation of the national command center (20 January 2020) helped with the diligent cooperation of citizens across the country, Taiwan has been praised as a success model in this pandemic and ranked third internationally for “COVID Resilience” by Bloomberg. At the time of the current study (September-October 2020), Taiwan with the population of 23,561,236, had 552 confirmed cases and 7 deaths. Due to the successful containment of the virus and very low risk community infection, Taiwan has escaped any forms of shutdown and daily life remained the “norm” except for wearing of masks and trace tracking [19]. The impressive record of success held with no interruption up to the time of writing this paper (19 April 2021), with 1076 confirmed cases and 11 deaths. Taiwan now has just started the vaccination program with 32,389 having the first dose.

Despite the international acclaimed success, medical staff are still under increased pressure to safeguard the nation through this evolving health crisis. Modeled after the National Health Service (NHS) of the UK, Taiwan boasts a world standard high-quality and low-cost national health insurance system (NHIS) that is accessible to every citizen. Historically, Taiwan’s hospitals have never shut down their services even in the height of war time (spring of 1945); except that during 2003’s SARS outbreak, Taiwan learned bitter experiences when several big hospitals collapsed due to serious in-hospital infection. Hence, protection of healthcare workers to secure hospital capacity has become top priority in 2020 COVID-19 pandemic. The central command center maintains steady personal protection equipment supply through national security stockpile program, quickly re-configures hospital workflow and ensures sufficient manpower supply for duty shift. Thus, throughout this pandemic, medical facilities remained operational, while the “COVID prevention mode” is enacted. Using our study organization (Taipei City Hospital, TPECH) as an example, the following changes were made by the hospital group. (1) All people entering the vicinity must wear medical masks at all time; (2) The designated personnel serving at the front door manage the admittance to patients and visitors, with justified reasons only. All entrants must have their temperatures taken and disinfect their hands; (3) If anyone has a fever, he/she must be transferred to the screening station immediately; (4) A screening station is set outside the emergency room. Emergency room staff take turns to man the station; (5) Any patients who have a fever when entering the hospital will be transferred to the fever screening stations for preliminary examination and treatment; (6) Use special passages and separate elevators to transport any suspected and confirmed COVID patients. Disinfect the used paths immediately after transportation; (7) Establish a special negative pressure isolation ward for COVID-19; (8) Assign specialized staff from each unit to care for suspected and confirmed COVID patients. Before entering the ward, the staff must wear a full set of protective clothing. More stringent restrictions were introduced such as a total ban of visitors, when the pandemic escalated intermittently. A way of obtaining valuable insight into the state of patient safety in the current unprecedented crisis situation, is thus to assess the prevailing patient safety culture and staff well-being in the frontline health institutions.

Objective: This study aims to evaluate the patient safety culture and employee well-being of a large hospital group in the capital metropolitan of Taiwan during the COVID-19 pandemic (data collection: September–October 2020). As a pulse survey, we systematically explore personal background factors as predictors of the patient safety culture and well-being. Furthermore, being the first study post-pandemic, we aim to gain insights on the impact of the crisis on both staff patient safety attitudes and well-being by comparing the COVID-19 cohort against the historic data in the same institution.

## 2. Materials and Methods

### 2.1. Study Design

This is a cross-sectional, survey-based study. Hospital staff who took part in any form of quality improvement activities, such as the quality control circle, symbol of national quality (SNQ, certification of safety and quality for health-related products and services in Taiwan), were invited to participate. Participants completed a self-administered questionnaire during the COVID-19 pandemic (2020). The same questionnaire was administered to all hospital employees in 2018 and provided the historic data as baseline.

All data are available in a deidentified format upon reasonable request to the Taipei City Hospital or the primary corresponding author.

### 2.2. Setting

Local institutional ethical approval was obtained from Taipei City Hospital Institutional Review Board (Permit number: TPECHIRB-10907001-E). Taipei City Hospital (TPECH) is a group of ten municipal general and specialty hospitals located in the metropolitan Taipei City, and is the largest healthcare organization in Northern Taiwan. The hospital group has 4.9 million patient contacts per year. About 4200 staff work in TPECH, with nearly 800 doctors, and more than 2000 nurses. Almost all staff are Taiwanese (ethnic Chinese).

### 2.3. Data Collection

Baseline data were collected using Google forms (Google LLC, Mountain View, CA, USA) between October to November 2018. Google forms were also used for the COVID-19 group between September to October 2020. In both instances, invitations were sent by email to potential participants. One reminder email was sent near the end of the set date for data collection. Questionnaires were anonymous to encourage participation and open responses.

### 2.4. Questionnaire Items

#### 2.4.1. Demographic Questions

The background information of the participants collected in both questionnaire surveys includes gender, age, job role, medical division, tenure at current position, education, managerial position, frequency of direct patient contact and reporting of any safety incidents in the past 12 months. For the purpose of analysis, staff were categorized into physicians, nurses and other hospital staff (including allied health professionals, pharmacists, administrative personnel and healthcare support workers). Medical divisions were re-grouped into critical care units (including ICU, operation rooms and emergency room), inpatient wards, ambulatory clinics and others (including administrative offices, auxiliary and support units).

#### 2.4.2. Safety Attitudes Questionnaire (SAQ)

The safety attitudes questionnaire (SAQ) assesses perceptions of safety across six dimensions: teamwork climate, safety climate, job satisfaction, stress recognition, perception of management and work conditions. It is based on the theoretical framework for analyzing risk and safety in clinical medicine, encompassing multilevel risk-related factors [5]. It is the most widely used tool to evaluate patient safety culture with good psychometric properties, and has been applied to various clinical areas such as general wards, emergency rooms, operating rooms and intensive care units [20,21,22]. The SAQ has also been validated for the Taiwanese medical setting and has been used to collect nationwide data since 2009 [6]. Exemplary items are “I have the support I need from other personnel to care for patients” (teamwork climate) and “I am encouraged by my colleagues to report any patient safety concerns I may have” (safety climate). The SAQ has 33 questions, using 5-point Likert scales from strongly disagree, disagree, neutral, agree to strongly agree. The higher the score, the more positive the attitude towards patient safety.

#### 2.4.3. Exhaustion

According to past research, emotional exhaustion is the core component of burnout compared with other dimensions (depersonalization and personal accomplishment) and the most obvious manifestation of the syndrome [23]. The 9-item emotional exhaustion scale from the Maslach burnout inventory (MBI) [24,25] was used to measure exhaustion (e.g., “I feel used up at the end of the workday”). The same 5-point Likert scales were used to keep consistency within the questionnaire (1 = strongly disagree to 5 = strongly agree). The original MBI scale uses a 7-point rating scale. However, researchers have used the 5-point rating scales for exhaustion, the same as the SAQ, and reported acceptable reliability [3] and validity [26]. For instance, using confirmatory factor analysis, Wu [26] found that the exhaustion scale had a CR of 0.91 for 965 hospital workers in Taiwan. In the present study, the internal consistency reliability α was 0.91. Although a lack of exhaustion is conceptualized as an indicator of employee well-being, to facilitate intuitive understanding of the results, we retain the original score to express the level of exhaustion. That is, the higher the score, the more experience of exhaustion.

### 2.5. Statistical Analysis

Group characteristics were demonstrated with simple descriptive statistics using Pearson chi-square tests to compare the distribution of characteristics among groups. ANOVA and independent sample *t* tests were used to compare SAQ and exhaustion across groups. Radar plots comparing SAQ scores among professions were created using RStudio v3.6.1.

For the COVID-19 cohort, linear regression of background variables was performed to identify which characteristics influence SAQ scores and the direction of any effect. All background variables in the linear regression analysis were statistically controlled in the hierarchical regression with patient safety culture predicting exhaustion. In all the analyses, statistical significance was set at *p* ≤ 0.05. Data were analyzed using SPSS v25.

## 3. Results

### 3.1. Sample Characteristics and Patient Safety Culture Survey Results

Among the 337 respondents in the 2020 sample, 284 (84.3%) were female and 197 (58.5%) were above the age of 40. Furthermore, 238 (70.6%) had a college degree and 111 (32.9%) held managerial positions. The heavy skew towards female and highly educated personnel is representative of the Taiwanese healthcare work force demography [6]. Almost all of the respondents had direct patient contact (93.2%), while most (59.3%) reported no patient safety incidents in the past year. The majority of respondents (67.7%) worked in their respective organizations for more than 10 years. In term of medical division, 39 (11.5%) worked in critical care units, 102 (30.3%) in inpatient wards and 83 (24.7%) in Ambulatory Clinics. Overall, physicians accounted for 27 (8%) of the respondents while nursing staff accounted for 192 (57%). Respondent characteristics are shown in Table 1, along with those of the baseline sample in 2018 for easy comparison.

Scores for all the six patient safety culture dimensions and employee well-being (reversely indexed as exhaustion) represented normal distributions with skewness and kurtosis between 1 and −1. Furthermore, internal consistency reliability was very good for all study variables, with Cronbach’s alphas between 0.89 (stress recognition) and 0.96 (job satisfaction). We grouped respondents by their medical roles into three professional groups, nurses (192), physicians (27) and others (118). There was statistically significant difference in overall mean score of SAQ (F(2, 2790) = 37.91, *p* < 0.001), and six of the SAQ dimensions between staff groups in the 2018 baseline sample. Nurses consistently reported the lowest scores on patient safety culture, trailing behind doctors and others. For example, overall SAQ scores for doctors and other clinical staff were 4.21 and 4.09 respectively, significantly higher than those for nurses (3.89). However, there was no statistically significant difference in overall mean score of SAQ (F(2, 271) = 1.02, *p* = 0.36), nor any of the SAQ dimensions between staff groups in 2020. For example, overall SAQ scores were 3.91, 4.06 and 3.94 for doctors, other clinical staff and nurses respectively, with no statistical difference among them. Scores for the six SAQ dimensions by professional group are shown in Figure 1.

### 3.2. Comparison

To facilitate comparison, we calculated positive response rates based on the commonly used scoring algorithm [6] in the SAQ literature. Namely, positive response rates are expressed as the proportion of respondents that “agree” (4) or “strongly agree” (5) with a given statement on a 5-point Likert scale. Vice versa for negatively worded items. The results showed that the positive response rates were higher than 50% for all the SAQ items in each dimension. Furthermore, the positive response rates were higher than 60% for all except for one item in stress recognition (“I am more likely to make errors in tense or hostile situations.”) and one in the working condition (“The levels of staffing in this clinical area are sufficient to handle the number of patients”). All items in the teamwork climate dimension reached higher than 70% agreement.

Following the common procedure in the SAQ literature and the recommendations of the local Joint Commission of Taiwan (JCT) [6], we converted the average of each SAQ variable into a percentile system of 0–100 points. The formula is the average score minus 1 then multiplied by 25. As the current Covid cohort is much smaller than the institutional baseline sample, we tested for equality of variances on the six SAQ dimensions, the overall SAQ score and exhaustion. The Levene’s test found no evidence of in equality for the six SAQ dimensions and the overall SAQ score (F = 0.577, 2.362, 0.885, 0.111, 0.045, 1.801, 1.050, ns). However, the equality assumption was not met for exhaustion (F = 26.974, *p* < 0.001), thus the adjusted t statistic is reported here. Table 2 shows that compared with the baseline sample in 2018, scores for working condition were significantly higher in the 2020 sample. However, stress recognition showed an opposite trend, with scores in 2018 significantly higher than those in 2020. There were no significant differences in the overall and other dimensions of SAQ scores. Scores for exhaustion were significantly higher in 2020 than in 2018.

### 3.3. Patient Safety Culture and Well-Being by Demography

We examined differences in patient safety culture and well-being by various demographical and job characteristics, including medical division, gender, age, managerial position, patient contact, job role, tenure and incident reporting behavior. To ensure sufficient subsample sizes for cross-group comparisons using ANOVA and *t*-tests, we re-grouped respondents by their age (below 40 years = 140, above 40 years = 197), job role (nurses = 192, physicians = 27, others = 118), medical division (critical care = 39, inpatient wards = 102, ambulatory clinics = 83, others = 113) and tenure (below 10 years = 109, above 10 years = 228). Results showed no statistically significant differences in any of the study variables by medical division, gender, patient contact or job role. Using the overall mean score of SAQ as an example, employees working in critical care units, inpatient wards, ambulatory clinics and other hospital units were not different (F(3270) = 1.65, *p* = 0.18). Male and female employees were not different (t(42) = 0.09, *p* = 0.93). Those with or without direct patient contact were not different (t(272) = −0.92, *p* = 0.36). Nurses, physicians and other hospital staff were not different (F(2271) = 1.02, *p* = 0.36).

However, older employees scored significantly higher than their younger counterparts on overall patient safety culture and the dimensions of teamwork climate, safety culture, job satisfaction and perception of management (show in Table 3). Managers scored significantly higher than non-managerial staff on overall patient safety culture and all its six dimensions (Table 4). Employees who reported safety incidents in the past 12 months scored significantly higher than those who did not report any incidents on overall patient safety culture and all its dimensions except for stress recognition (Table 5). The same pattern was observed for those worked for longer than 10 years in their organizations (Table 6). Overall, the differences on the endorsement of patient safety culture were most consistent and pronounced between managers and non-managerial staff (Table 4). Therefore, it is suggested that any future attempts of improvements on patient safety culture should engage the entire organization, with more resource deployment and stronger support for the non-managerial staff. Finally, it is worth pointing out that there were no statistically significant differences in the exhaustion scores in any of the comparisons. Thus, the level of exhaustion (lack of well-being) remained constant across all groups of hospital staff.

### 3.4. Predicting Patient Safety Culture and Well-Being

Table 7 shows the correlation matrix of all the variables for multivariate analysis. Among demographics and job characteristics, higher education and being a manager consistently positively correlated with patient safety culture. Older age correlated positively with teamwork climate, safety climate, job satisfaction and perception of management. Longer tenure and having reported incidents significantly correlated with all but the stress recognition dimension of the patient safety culture. Gender and direct patient contact did not significantly correlate with patient safety culture. None of the personal and job characteristics significantly correlate with exhaustion. The pattern of bivariate correlations between personal/job factors and patient safety culture corroborates the above mentioned results of cross-group comparisons (Table 3, Table 4, Table 5 and Table 6). Among the dependent variables, all six SAQ dimensions significantly positively correlated with one another. Thus, following the practice in the SAQ literature, we examined each dimension of and the overall patient safety culture in a later analysis. However, it is worth pointing out that stress recognition consistently had the weakest correlations with all the other five aspects of SAQ (r = 0.23–0.33), and was the only dimension that correlated positively with exhaustion. Although this is in line with previous findings in the SAQ research [5], caution is needed in interpreting results pertaining to stress recognition.

Following the bivariate analysis, we further explored personal demographic and job characteristics as predictors of staff perceptions of patient safety culture and their well-being, using multivariate analysis. As scores on all the dimensions of patient safety culture and exhaustion conformed to normal distribution, we conducted eight separate multiple linear regressions in search for significant predictors, and results are shown in Table 8. Having all the demographic and job variables competing against one another, the results of multivariate analysis generally corroborated findings of bivariate correlations described above. Namely, holding a managerial position and having longer tenure were salient predictors of various dimensions of and the overall SAQ. Specifically, positive attitudes towards teamwork climate, safety climate and working condition were more predictable by employees’ tenure, while all dimensions of the patient safety culture except stress recognition could be predicted by holding a managerial position. Furthermore, the significant effect of managerial position persisted in predicting the overall score of patient safety culture (*β* = 0.21, *p* < 0.01). In addition, being female could predict higher scores on perception of management, and having direct patient contact could predict higher scores on job satisfaction. Again however, exhaustion could not be predicted by any of the employees’ background variables.

We then examined the predictive power of each dimension of the patient safety culture and the overall SAQ score on employees’ exhaustion. We tested eight hierarchical regression models and results are shown in Table 9. We treated the seven personal background variables as control variables (baseline model). Having controlled for their effects, we then entered one specific SAQ dimension in each research model (model 1 to model 6). For example, model 1 shows the predictive power of teamwork climate on exhaustion after taken out the effects of personal background variables. In model 7, the overall SAQ score was entered after controlling for the background variables. In model 8, all six SAQ dimensions were entered together to compete in predicting exhaustion. Results in Table 9 show that after controlling for personal factors, teamwork climate, safety climate, job satisfaction and stress recognition were all significant predictors of employees’ exhaustion. Specifically, teamwork climate increased 2% of variation in explaining exhaustion (model 1, F(1259) = 4.91, *p* < 0.05), safety climate increased 2% (model 2, F(1259) = 6.51, *p* < 0.05), job satisfaction 2% (model 3, F(1259) = 4.76, *p* < 0.05) and stress recognition 11% (model 4, F(1259) = 32.62, *p* < 0.001). Finally, model 8 shows that all six dimensions of SAQ together could increase 19% of variation in explaining exhaustion, F(6, 254) = 10.36, *p* < 0.001. More importantly, after controlling for all the personal factors and competing against all other SAQ dimensions, safety climate (*β* = −0.32, *p* < 0.05) and stress recognition (*β* = 0.44, *p* < 0.001) were still significant predictors for employees’ exhaustion. Note however, with higher scores on stress recognition, exhaustion was higher.

## 4. Discussion

This study investigated the patient safety culture and hospital staff well-being in Taiwan during the COVID-19 pandemic and compares the COVID-19 cohort against the historical 2018 data in the same hospital group. The result shows that comparing to the baseline in 2018, staff reported higher (more positive) patient safety attitudes pertaining to the working condition during COVID-19. However, the COVID-19 cohort did suffer greater exhaustion compared to their counterparts before the pandemic. In multivariate analysis, we found that holding a managerial position was the most salient predictor of SAQ (overall score and 5/6 subscale scores). Experiences at work could also predict three SAQ subscale scores. Furthermore, the results of multivariate analysis show that after controlling for all the personal factors, SAQ dimensions of teamwork climate, safety climate, job satisfaction and stress recognition could each predict staff exhaustion. The negative relation between safety climate and exhaustion and the positive relation between stress recognition and exhaustion remained significant even when all facets of SAQ were jointly considered. Finally, none of the personal background factors were related to staff exhaustion.

### 4.1. Theoretical Implications

Our findings have several important theoretical implications. First and foremost, we contribute to the literature by investigating the patient safety culture amid an unprecedented global health crisis, the coronavirus pandemic. Our pulse survey of staff in the largest metropolitan hospital group in Taiwan reveals that the positive attitudes for working condition of the patient safety culture were improved during COVID-19 compared to the 2018 historic data in the same organization. The improvement of patient safety culture may be attributable to the heightened awareness of safety and risk saliency (“COVID effect”) brought about by the persisting pandemic, more potently among the frontline healthcare professionals. In a risk aversive culture such as Taiwan, when facing such an unknown novel disease, everyone quickly switched on a “protective mode” characterized with heightened safety awareness, more attention to risks and taking extra precautions, such as wearing a complete set of protective apparel, frequent hand washing and disinfection of items with 75% alcohol at any time. The perception of better working condition may also be the result of the nationwide resource mobilization and increased investment in the health system, for example extra human resources, operation funds and recognition compensation schemes for directly involved personnel. Finally, our observed increase in SAQ scores (except for stress recognition) also mirror the continuous long-term trend of improvement of patient safety culture in Taiwan [6]. However, we need to be cautious in interpreting the comparison results as the COVID cohort is different from the 2018 institutional sample. The participants of this study were employees who have been involved with quality improvement activities. They would have paid more attention to patient safety. In addition, there were more managers, older and more experienced staff in the COVID cohort. They have been found to have a higher awareness of patient safety [27].

Second, related to the above, the finding that managers reported higher scores on overall patient safety culture and all its dimensions except for stress recognition may be explained by the fact that the managerial position ensue more responsibilities and stronger identification toward the organization goals [3]. Another possible explanation is that managers are viewed as the key persons in fostering the institutional patient safety culture, thus often targeted in training workshops, lectures, policy revisions and awareness campaigns by the JCT and the Ministry of Health and Welfare (MoHW) in Taiwan [6]. For example, the JCT holds annual classes to develop patient safety culture in hospitals, with personnel from 103 institutions (mainly managers and team leaders) joining the 2016 training courses on Team Resource Management and caregiver resilience [6]. We also found that positive attitudes towards teamwork climate, safety climate and working condition were more predictable by employees’ tenure. Tenure does correlate with holding a managerial position (r = 0.32 in this study). Smits and colleagues [27] also found that working experience was positively related to higher teamwork climate and communication openness. To sum, managers and more experienced staff seem to endow strongly with patient safety culture.

Third, we extend the SAQ literature by including staff burnout as an outcome variable of the patient safety culture. Teamwork climate, safety climate and job satisfaction each significantly predicted lower exhaustion while stress recognition significantly predicted higher exhaustion. We also found that safety climate and stress recognition can most effectively predict staff exhaustion after controlling for personal variables and competing with other dimensions of SAQ. In line with the conservation of resources (COR) theory, those with greater resources are less vulnerable to resource loss and more capable of resource gain [28]. The staff who possess more team resources (higher teamwork climate) and more personal resources (higher job satisfaction) are less vulnerable to burnout. Although working condition was not a significant predictor of exhaustion in this study, previous research did find that good work environments could attenuate the relationship between nurses’ burnout and patient outcomes [12]. The relationship between working condition and staff exhaustion deserves further exploration. While patient safety culture in general could reduce staff exhaustion, it is surprising to note that stress recognition increased exhaustion. A possible explanation was that staff who are already strained have less energy to reflect and exercise self-care. In the COVID-19 scenario, the increased workload and pressure obligate employees to focus on immediate problems, leaving them with no time to reflect on their own mental state. This diminished self-awareness may affect patient safety in the longer run. This speculation is consistent with COR’s proposition that when people’s resources are outstretched or exhausted, they enter a defensive mode to preserve the self, which is often defensive, aggressive and may become irrational [28]. This unfortunately sets motion the resource loss spiral, which incur even greater strain. While the positive relationship between stress recognition and exhaustion needs to be clarified, it is however beyond the scope of the cross-sectional study and must be tested by longitudinal data.

However, comparing to the 2018 institutional baseline, we found that employees reported higher levels of exhaustion during the COVID-19 pandemic. This is contrary to what was found in a recent study, that is when more attention was paid to patient safety culture, the level of employee exhaustion was lower [29]. Note that our participants were employees actively involved in quality improvement activities. To reconcile the inconsistency, we need to look closer at the context of our present study, the evolving pandemic. Due to the COVID-19 pandemic, the Taiwanese government has implemented several effective measures to protect the nation’s health system, including distinguishing the suspected and confirmed patients, preventing all patients from staying except in only a few hospitals and educating the public to reduce unnecessary medical treatments. Moreover, annual hospital accreditation is postponed, allowing employees to devote more effort to the healthcare activities. Despite these interventions, many COVID-19-related measures have increased workload for the health workers. Working in the first line of defense against the pandemic, anxieties over the ignorance of the disease, fear of infection and transmission to family and friends could lead to the up surging level of staff exhaustion [17].

Finally, it is worth pointing out that we found the stress recognition had the lowest scores in the SAQ, and showed a different pattern of relationship with exhaustion. The SAQ literature has documented some anomaly concerning this particular aspect of patient safety, for instance, stress recognition usually has the lowest correlation with other SAQ aspects [5]. Studies in Taiwan has also noted problems with the stress recognition dimension [26].

### 4.2. Managerial Implications

Our findings have relevant implications for practitioners. First, our results highlight that managers generally have higher scores on patient safety culture. As resources are limited, it is advisable that non-managerial employees should be targeted for training in the future. For instance, managers who participated the annual classes of JCT can hold workshops to share the knowledge of patient safety culture for their work units, especially for nurses who usually have lower patient safety culture and higher exhaustion. Second, our findings suggest that teamwork climate, safety climate and job satisfaction have individually negative significant predictions on exhaustion. Therefore, these three aspects should be targeted and strengthened in interventions aimed to enhance the well-being and resilience of employees. However, it is important to note that many medical accidents and adverse incidents are not simply individual’s errors but the latent failure of the organization and system [7,8], so creating a good working environment is the priority. Finally, we found that staff exhaustion increased during the pandemic across the board. Existing studies have shown that employee burnout is related to patient mortality and hospital stay [12]. Hence, hospitals should consider increasing the number of caregivers, giving longer rest periods, providing resting places and better protective equipment and so on to reduce employee burnout [16].

### 4.3. Limitation and Future Research Directions

The current study is subject to some limitations. First, our study design used purposive sampling to select participants, thus our study sample is not representative of the entire hospital group. Nonetheless, we systematically examined sample variations in demographics and job characteristics between the COVID-19 cohort and the organizational historic data. We also adopted multivariate analysis to account for any effects of personal factors in predicting SAQ and exhaustion. The evidence of the myriad impact of the pandemic on patient safety and staff well-being will be more convincing if future studies can recruit larger and representative samples of hospital workers. A second limitation is that we used self-report measures, which may increase the threat of common method variance (CMV) bias [30]. While it is valuable to gain timely insights on the impact of the evolving pandemic, our findings are better read as an impulse survey. To get more comprehensive knowledge, we suggest future studies should consider including objective measurements of staff performance and patient outcomes. Third, our data are cross-sectional, thus no causal inferences should be made pertaining to the relationships among study variables. As almost all the existing studies on patient safety employed a cross-sectional design, we are unable to generate comprehensive knowledge on the prospective effects of patient safety on staff performance and well-being [2,6]. Future studies should adopt longitudinal study designs to establish the casual order of patient safety and staff well-being. The relationship may even be reciprocal when the temporal effects could be modeled. Finally, we focused on exhaustion as the only indicator of staff well-being. Although most SAQ studies did not include staff outcome variables, it is a significant lacking, especially as well-being is going mainstream in the post-pandemic era. Future studies should include more varied indicators of staff well-being (such as health and work–life balance) and potential systemic protectors of staff well-being, such as support [31] and positive work environment [12].

## 5. Conclusions

To conclude, our study is among the few investigating safety culture during the COVID-19 pandemic [31] and the first in Taiwan. We offered important insight on the state of institutional patient safety culture and staff well-being under unprecedented crisis circumstances. Our results suggest that on the one hand, the COVID pandemic has triggered the nationwide crisis management mode and have resulted in more resources being mobilized and deployed in the health system leading to hospital staff across medical roles to endorse higher positive attitudes towards the working condition aspect of the patient safety culture. On the other hand, the burden of coping with the pandemic and responsibilities of protecting the health of the nation have also caused an increase on staff exhaustion. Within the COVID-19 cohort, managerial role was associated with higher SAQ scores. SAQ dimensions of teamwork climate, safety climate and job satisfaction were associated with lower exhaustion, while stress recognition was associated with higher exhaustion. These findings suggest that healthcare institutions may use the pandemic as a catalyst for hospital-wide patient safety and quality improvement interventions, targeting the front-line staff. Meanwhile, in the post-pandemic spirit of compassion and value for well-being, healthcare institutions should also endeavor to cultivate a positive work environment to alleviate staff exhaustion. Never waste a disaster.

## Figures and Tables

**Figure 1 ijerph-18-04537-f001:**
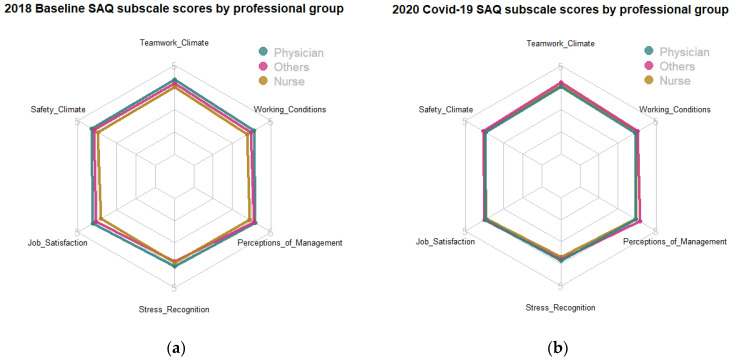
Radar plot: (**a**) 2018 baseline SAQ subscale scores by the professional group and (**b**) 2020 COVID-19 SAQ subscale scores by the professional group.

**Table 1 ijerph-18-04537-t001:** Respondent characteristics.

Variable	2018 SAQ	2020 SAQ	Chi-Square/*p*-Value
	*n* (%)	*n* (%)	
Medical Division			
Critical care units	515 (15.9%)	39 (11.5%)	6.99/0.07
Inpatient wards	1048 (32.4%)	102 (30.3%)	
Ambulatory clinics	673 (20.8%)	83 (24.7%)	
Others	996 (30.8%)	113 (33.5%)	
Gender			
Female	2701 (83.6%)	284 (84.3%)	0.06/0.80
Male	531 (16.4%)	53 (15.7%)	
Age			
≤40	1941 (60.1%)	140 (41.5%)	42.27/<0.001
>40	1291 (39.9%)	197 (58.5%)	
Managerial Position			
No	2947 (91.2%)	226 (67.1%)	177.54/< 0.001
Yes	285 (8.8%)	111 (32.9%)	
Incident Reports			
None	2297 (71.7%)	200 (59.3%)	19.41/<0.001
At least one	935 (28.9%)	137 (40.7%)	
Job Role			
Physician	242 (7.5%)	27 (8%)	2.42/0.30
Nurse	1718 (53.2%)	192 (57%)	
Others	1272 (39.4%)	118 (35%)	
Tenure			
≤10 years	2045 (63.3%)	109 (32.3%)	120.72/<0.001
>10 years	1187 (36.7%)	228 (67.7%)	
Education			
No high school degree	31 (1.0%)	1 (0.3%)	Not available
High school degree	278 (8.6%)	4 (1.2%)	
College degree	2572 (79.6%)	238 (70.6%)	
Graduate degree	351 (10.9%)	94 (27.9%)	
Patient Contact			
No	278 (8.6%)	23 (6.8%)	1.03/0.31
Yes	2954 (91.4%)	314 (93.2%)	
Total	3232 (100%)	337 (100%)	

**Table 2 ijerph-18-04537-t002:** Changes in patient safety culture and exhaustion, 2018 vs. 2020.

	2018 Baseline		2020 SAQ		*t*	*p*
	N	Mean (SD)	N	Mean (SD)		
Teamwork Climate	2876	77.51(19.49)	302	78.55(18.70)	−0.88	0.377
Safety Climate	3010	76.75(19.54)	318	75.45(18.48)	1.13	0.258
Job satisfaction	3226	73.40(22.74)	331	73.69(22.05)	−0.22	0.825
Stress Recognition	3204	71.78(23.16)	320	68.54(22.34)	2.40	0.017
Perception of Management	3145	74.97(21.33)	328	75.06(21.20)	−0.07	0.944
Working Condition	2926	71.48(21.89)	304	74.79(21.00)	−2.52	0.012
Patient Safety Culture	2793	74.60(17.71)	274	74.30(17.06)	0.28	0.782
Exhaustion	3185	36.99(23.74)	313	47.67(20.30)	−8.73	<0.001

**Table 3 ijerph-18-04537-t003:** Differences between means by age.

	≤40 Years Old		>40 Years Old		*t*	*p*
	Mean	SD	Mean	SD		
Teamwork Climate	4.00	0.81	4.24	0.69	−2.73	0.007 **
Safety Climate	3.90	0.79	4.10	0.69	−2.40	0.017 *
Job Satisfaction	3.81	0.92	4.05	0.84	−2.45	0.015 *
Stress Recognition	3.73	0.97	3.75	0.84	−0.20	0.839
Perception of Management	3.88	0.89	4.09	0.81	−2.20	0.029 *
Working Condition	3.90	0.89	4.06	0.80	−1.63	0.103
Patient Safety Culture	3.87	0.74	4.05	0.64	−2.16	0.032 *
Exhaustion	2.95	0.80	2.88	0.82	0.75	0.455

* *p* < 0.05, ** *p* < 0.01.

**Table 4 ijerph-18-04537-t004:** Differences between means by managerial position.

	Non-Managers		Mangers		*t*	*p*
	Mean	SD	Mean	SD		
Teamwork Climate	3.99	0.75	4.44	0.64	−5.03	<0.001 ***
Safety Climate	3.86	0.73	4.33	0.66	−5.57	<0.001 ***
Job Satisfaction	3.77	0.89	4.30	0.76	−5.29	<0.001 ***
Stress Recognition	3.67	0.87	3.89	0.93	−2.09	0.037 *
Perception of Management	3.86	0.86	4.30	0.75	−4.69	<0.001 ***
Working Condition	3.84	0.83	4.29	0.78	−4.51	<0.001 ***
Patient Safety Culture	3.83	0.67	4.26	0.62	−5.11	<0.001 ***
Exhaustion	2.94	0.74	2.84	0.95	0.97	0.333

* *p* < 0.05, *** *p* < 0.001.

**Table 5 ijerph-18-04537-t005:** Differences between means by incident reporting.

	No Incident Reporting		At Least One Incident Reporting		*t*	*p*
	Mean	SD	Mean	SD		
Teamwork Climate	4.02	0.76	4.30	0.71	−3.22	0.001 **
Safety Climate	3.89	0.74	4.19	0.71	−3.58	<0.001 ***
Job Satisfaction	3.82	0.91	4.14	0.80	−3.39	0.001 **
Stress Recognition	3.66	0.88	3.85	0.91	−1.87	0.062
Perception of Management	3.88	0.84	4.18	0.82	−3.22	0.001 **
Working Condition	3.87	0.83	4.15	0.83	−2.91	0.004 **
Patient Safety Culture	3.84	0.66	4.14	0.67	−3.70	<0.001 ***
Exhaustion	2.92	0.78	2.89	0.85	0.34	0.732

** *p* < 0.01, *** *p* < 0.001.

**Table 6 ijerph-18-04537-t006:** Differences between means by tenure.

	≤10 Years		>10 Years		*t*	*p*
	Mean	SD	Mean	SD		
Teamwork Climate	3.93	0.80	4.25	0.70	−3.62	<0.001 ***
Safety Climate	3.80	0.78	4.13	0.70	−3.79	<0.001 ***
Job Satisfaction	3.72	0.91	4.06	0.85	−3.35	0.001 **
Stress Recognition	3.82	0.82	3.70	0.93	1.05	0.292
Perception of Management	3.84	0.87	4.08	0.83	−2.48	0.013 *
Working Condition	3.74	0.87	4.11	0.80	−3.55	<0.001 ***
Patient Safety Culture	3.79	0.72	4.06	0.65	−3.00	0.003 **
Exhaustion	2.97	0.71	2.88	0.86	0.98	0.327

* *p* < 0.05, ** *p* < 0.01, *** *p* < 0.001.

**Table 7 ijerph-18-04537-t007:** Means, standard deviations and correlations among the study variables.

		Mean	SD	1	2	3	4	5	6	7	8	9	10	11	12	13	14	15
1.	Gender																	
2.	Age			−0.08														
3.	Managerial Position			0.11 *	0.40 **													
4.	Incident Reporting			−0.01	0.13 *	0.45 **												
5.	Tenure			−0.17 **	0.54 **	0.32 **	0.13 *											
6.	Education Years			0.15 **	0.10	0.38 **	0.10	0.06										
7.	Patient Contact			−0.01	−0.11 *	0.01	0.10	−0.06	0.03									
8.	Teamwork Climate	4.14	0.75	0.03	0.16 **	0.28 **	0.18 **	0.20 **	0.14 *	0.09	(0.89)							
9.	Safety Climate	4.02	0.74	<0.01	0.14 *	0.30 **	0.20 **	0.21 **	0.21 **	0.02	0.89 **	(0.89)						
10.	Job Satisfaction	3.95	0.88	−0.01	0.13 *	0.28 **	0.18 **	0.18 **	0.12 *	0.11	0.78 **	0.83 **	(0.96)					
11.	Stress Recognition	3.74	0.89	0.08	0.01	0.12 *	0.10	−0.06	0.15 **	0.01	0.30 **	0.33 **	0.23 **	(0.89)				
12.	Perception of Management	4.00	0.85	−0.06	0.12 *	0.24 **	0.18 **	0.14 *	0.15 **	0.02	0.79 **	0.82 **	0.78 **	0.26 **	(0.90)			
13.	Working Condition	3.99	0.84	−0.05	0.09	0.25 **	0.16 **	0.20 **	0.18 **	<0.01	0.77 **	0.81 **	0.77 **	0.29 **	0.88 **	(0.91)		
14.	Patient Safety Culture (Overall)	3.97	0.68	−0.01	0.13 *	0.30 **	0.22 **	0.18 **	0.20 **	0.06	0.90 **	0.93 **	0.88 **	0.52 **	0.91 **	0.90 **	(0.91)	
15.	Exhaustion	2.91	0.81	−0.02	−0.04	−0.06	−0.02	−0.05	−0.05	<0.01	−0.18 **	−0.21 **	−0.19 **	0.31 **	−0.17 **	−0.16 **	−0.08	(0.91)

Note: Scale reliabilities (Cronbach’s alphas) appear in brackets on the diagonal. Gender (female = 0, male = 1); age (≤40 = 0, >40 = 1); managerial position (no = 0, yes = 1); incident reporting (none = 0, at least one = 1); tenure (≤10 years = 0, >10 years = 1); patient contact (no = 0, yes = 1) * *p* < 0.05, ** *p* < 0.01.

**Table 8 ijerph-18-04537-t008:** Linear regression of background variables.

	Teamwork Climate			Safety Climate			Job Satisfaction			Stress Recognition			Perception of Management			Working Condition			Patient Safety Culture			Exhaustion		
	*B*	*SE B*	*β*	*B*	*SE B*	*β*	*B*	*SE B*	*β*	*B*	*SE B*	*β*	*B*	*SE B*	*β*	*B*	*SE B*	*β*	*B*	*SE B*	*β*	*B*	*SE B*	*β*
Medical Division_ Inpatient Ward	0.05	0.14	0.03	0.05	0.13	0.03	0.12	0.16	0.06	−0.09	0.17	−0.04	0.11	0.16	0.06	0.13	0.16	0.07	0.03	0.13	0.02	0.13	0.16	0.07
Medical Division_ Ambulatory Clinic	0.31	0.21	0.17	0.31	0.19	0.18	0.18	0.23	0.09	−0.07	0.25	−0.03	0.24	0.22	0.12	0.18	0.24	0.09	0.16	0.20	0.10	−0.22	0.23	−0.12
Medical Division_ Others	0.04	0.14	0.02	−0.03	0.14	−0.02	0.05	0.16	0.03	−0.14	0.18	−0.07	0.07	0.16	0.04	−0.05	0.16	−0.03	−0.04	0.13	−0.03	0.04	0.16	0.02
Gender	0.03	0.14	0.01	−0.11	0.13	−0.05	−0.10	0.15	−0.04	0.05	0.16	0.02	−0.29 *	0.14	−0.13 *	−0.17	0.16	−0.07	−0.08	0.13	−0.04	−0.02	0.15	−0.01
Age	0.08	0.11	0.05	<.01	0.10	<0.01	0.02	0.12	0.01	0.05	0.13	0.03	0.05	0.12	0.03	−0.08	0.12	−0.05	0.02	0.10	0.01	−0.03	0.12	−0.02
Managerial Position	0.33 **	0.12	0.21 **	0.36 **	0.11	0.23 **	0.45 ***	0.13	0.24 ***	0.14	0.14	0.07	0.36 **	0.13	0.20 **	0.35 *	0.14	0.20 *	0.30 **	0.11	0.21 **	−0.10	0.13	−0.06
Incident Reports	0.06	0.10	0.04	0.08	0.09	0.06	0.09	0.11	0.05	0.14	0.12	0.07	0.15	0.11	0.09	0.07	0.11	0.04	0.11	0.09	0.08	0.01	0.11	0.01
Job Role_ Nurse	0.24	0.18	0.15	0.16	0.17	0.11	0.07	0.21	0.04	<.01	0.23	<0.01	0.01	0.19	0.01	0.19	0.20	0.11	0.12	0.17	0.08	−0.14	0.21	−0.09
Job Role_ Others	0.29	0.22	0.18	0.16	0.21	0.10	0.26	0.24	0.14	0.09	0.26	0.05	0.28	0.23	0.16	0.27	0.25	0.15	0.24	0.21	0.16	−0.07	0.24	−0.04
Tenure	0.23 *	0.11	0.15 *	0.22 *	0.10	0.14 *	0.23	0.12	0.12	−0.19	0.13	−0.10	0.11	0.12	0.06	0.27 *	0.12	0.15 *	0.17	0.10	0.11	−0.09	0.12	−0.05
Education Years	0.02	0.04	0.03	0.07	0.04	0.11	<0.01	0.05	<0.01	0.09	0.05	0.11	0.04	0.05	0.05	0.08	0.05	0.11	0.06	0.04	0.09	−0.01	0.05	−0.01
Patient Contact	0.35	0.21	0.10	0.05	0.18	0.02	0.42 *	0.20	0.12 *	<0.01	0.21	<0.01	0.07	0.20	0.02	0.06	0.24	0.01	0.27	0.21	0.07	−0.01	0.20	<0.01
*R* ^2^	0.14			0.16			0.13			0.05			0.12			0.12			0.15			0.03		
*Adj R* ^2^	0.11			0.13			0.09			0.01			0.09			0.09			0.11			−0.01		
*F*	4.06 ***			4.78 ***			3.79 ***			1.23			3.65 ***			3.41 ***			3.79 ***			0.68		
*df*	(12, 289)			(12, 305)			(12, 318)			(12, 307)			(12, 315)			(12, 291)			(12, 261)			(12, 300)		

Note: Medical division (ref = critical care units); gender (female = 0, male = 1); age (≤40 = 0, >40 = 1); managerial position (no = 0, yes = 1); incident reports (none = 0, at least one = 1); job role (ref = physician); tenure (≤10 years = 0, >10 years = 1); patient contact (no = 0, yes = 1); * *p* < 0.05 ** *p* < 0.01 *** *p* < 0.001.

**Table 9 ijerph-18-04537-t009:** Hierarchical regression of patient safety culture on exhaustion (*n* = 268).

	Baseline Model		Model 1		Model 2		Model 3		Model 4		Model 5		Model 6		Model 7		Model 8	
	*β*	*T*	*β*	*t*	*β*	*t*	*β*	*t*	*β*	*t*	*β*	*t*	*β*	*t*	*β*	*t*	*β*	*t*
Step 1: Control Variables																		
Gender	−0.08	−1.22	−0.08	−1.18	−0.08	−1.28	−0.08	−1.27	−0.10	−1.66	−0.09	−1.36	−0.08	−1.25	−0.08	−1.22	−0.11	−1.80
Age	0.02	0.31	0.02	0.31	0.02	0.23	0.02	0.28	0.02	0.23	0.03	0.36	0.02	0.21	0.02	0.30	<0.01	0.00
Managerial Position	−0.09	−1.11	−0.07	−0.83	−0.07	−0.82	−0.06	−0.76	−0.11	−1.39	−0.07	−0.90	−0.08	−0.92	−0.08	−0.99	−0.07	−0.87
Incident Reports	0.02	0.22	0.02	0.31	0.03	0.41	0.02	0.33	−0.01	−0.18	0.03	0.36	0.02	0.34	0.02	0.28	<0.01	0.03
Tenure	−0.06	−0.85	−0.04	−0.57	−0.04	−0.53	−0.05	−0.67	−0.04	−0.58	−0.06	−0.84	−0.04	−0.59	−0.06	−0.78	0.02	0.29
Educational Year	−0.02	−0.24	−0.01	−0.11	0.01	0.11	−0.01	−0.10	−0.04	−0.58	0.00	−0.05	0.00	−0.05	−0.01	−0.15	<0.01	−0.02
Patient Contact	0.02	0.29	0.03	0.52	0.03	0.43	0.02	0.32	0.02	0.38	0.02	0.32	0.02	0.37	0.02	0.33	0.04	0.77
Step 2: Independent Variables																		
Teamwork Climate			−0.14	−2.22 *													−0.05	−0.41
Safety Climate					−0.17	−2.55 *											−0.32	−2.12 *
Job Satisfaction							−0.14	−2.18 *									−0.03	−0.29
Stress Recognition									0.34	5.71 ***							0.44	7.31 ***
Perception of Management											−0.11	−1.67					0.07	0.51
Working Condition													−0.11	−1.73			0.02	0.14
Patient Safety Culture (Overall)															−0.05	−0.80		
*Adj R* ^2^	−0.01		0.01		0.01		0.01		0.10		<0.01		<0.01		−0.01		0.17	
∆*R*^2^	0.02		0.02		0.02		0.02		0.11		0.01		0.01		0.00		0.19	
*F*	0.73		1.26		1.47		1.25		4.80 ***		0.99		1.02		0.72		5.26 ***	
∆*F*	0.73		4.91 *		6.51 *		4.76 *		32.62 ***		2.78		2.98		0.64		10.36 ***	
*(df)*	(7, 260)		(8, 259)		(8, 259)		(8, 259)		(8, 259)		(8, 259)		(8, 259)		(8, 259)		(13, 254)	

Note: Gender (female = 0, male = 1); age (≤40 = 0, >40 = 1); managerial position (no = 0, yes = 1); incident reports (none = 0, at least one = 1); tenure (≤10 years = 0, >10 years = 1); patient contact (no = 0, yes = 1); * *p* < 0.05 *** *p* < 0.001.

## Data Availability

The data that support the findings of this study are available upon reasonable request to the Taipei City Hospital or the primary corresponding author.

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
