# Peer review of "Post-Pandemic Patient Safety Culture: A Case from a Large Metropolitan Hospital Group in Taiwan"

_ijerph, 2021, doi:10.3390/ijerph18094537_

Round 1
Reviewer 1 Report
This is an interesting manuscript on patient safety culture and staff well-being admit the COVID-19 pandemic. It has several streghts, such as measures in different moments. However, I have some concerns before acceptance:
- More information on the country reality and the sector
- provide Ethical approval
- Update your introduction with current literature on Covid-19 reality. I suggest:
Greenberg, N., Docherty, M., Gnanapragasam, S., & Wessely, S. (2020). Managing mental health challenges faced by healthcare workers during covid-19 pandemic. bmj, 368.
Murphy, M., & Moret-Tatay, C. (2021). Personality and Attitudes Confronting Death Awareness During the COVID-19 Outbreak in Italy and Spain. Frontiers in Psychiatry, 12. https://doi.org/10.3389/fpsyt.2021.627018
Vindegaard, N., & Benros, M. E. (2020). COVID-19 pandemic and mental health consequences: Systematic review of the current evidence. Brain, behavior, and immunity, 89, 531-542.
4. Please, specify if assumptions were reached to carried out the statistical methods employed, as different sample size are involved. Also include effect sizes.
- Minor Typos
pag 408. styles of citation mixed. (Yu et al., 2020)
Author Response
Revisions made following Reviewer 1’s suggestions
- More information on the country reality and the sector
Response: In the last three paragraphs of the Introduction (preceding “Objective”), I have now provided a detailed description of the country reality charting key actions the Taiwanese government has taken since the outbreak of the crisis. I have also added COVID statistics at the time of study and that of writing up the paper. I have moved the paragraph describing the COVID-related changes in hospitals from “2.2 Setting” forward to give readers a snapshot of the medical sector.
- Provide Ethical approval
Response: This information is provided in 2.2, reading “Local institutional ethical approval was obtained from Taipei City Hospital Institutional Review Board (TPECHIRB-10907001-E).” I have now highlighted the statement in red.
- Update your introduction with current literature on Covid-19 reality. I suggest:
Greenberg, N., Docherty, M., Gnanapragasam, S., & Wessely, S. (2020). Managing mental health challenges faced by healthcare workers during covid-19 pandemic. bmj, 368.
Murphy, M., & Moret-Tatay, C. (2021). Personality and Attitudes Confronting Death Awareness During the COVID-19 Outbreak in Italy and Spain. Frontiers in Psychiatry, 12. https://doi.org/10.3389/fpsyt.2021.627018
Vindegaard, N., & Benros, M. E. (2020). COVID-19 pandemic and mental health consequences: Systematic review of the current evidence. Brain, behavior, and immunity, 89, 531-542.
Response: I have now update the introduction with current literature on Covid-19 reality, incorporating the findings and ideas from Greenberg and colleagues (2020), as well as Vindegaard, and Benros (2020). However, the Murphy and Moret-Tatay (2021) study does not seem closely relevant to our research interest.
- Please, specify if assumptions were reached to carry out the statistical methods employed, as different sample size are involved. Also include effect sizes.
Response: I have added the results of Levenel’s test for equality of variances between the two samples. The assumptions were reached for SAQ to carry out the statistical methods employed, but not for exhaustion.
- Minor Typos
Line 408. styles of citation mixed. (Yu et al., 2020)
Response: This inconsistency has now been corrected.

Reviewer 2 Report
The manuscript presents the results of an interesting study with practical implications for the management of hospital human resources. The authors present their work rigorously and well supported by previous literature. I believe that the manuscript can be published although I have found some errors that must be solved:
- Line 117. What is SNQ?
- Line 173: “strongly disagree” appears twice
- Add information on the reliability of the instruments used in your sample, especially regarding the measurement of emotional exhaustion, since the authors have modified the original evaluation scale (from seven points) to one of 5 points. Although this information appears much later (in table 7), it would be recommended that it appear in the description of the instruments used.
- In figure 1 the meaning of the initials TC, WC, PM, etc. should be added in the title or footer. The same in the tables (please, avoid using initials)
- Table 7 shows mean values and standard deviations of qualitative variables !!!! Like gender, age group, managerial position, etc. This is a very serious error; with this type of variables it does not make sense to calculate these statistics. Remove these statistics
- The reasoning used to perform the hierarchical regression analyzes is not well understood and Table 9 is confusing. The authors should clarify how the independent variables have been introduced into the model
Author Response
Revisions made following Reviewer 2’s suggestions
- Line 117. What is SNQ?
Response: SNQ refers to the Symbol of National Quality, which is the certification of safety and quality for health-related products and services in Taiwan, endorsed by the government. I have now added the full-term with explanatory wordings in the text.
- Line 173: “strongly disagree” appears twice
Response: This was a typo, and has been corrected.
- Add information on the reliability of the instruments used in your sample, especially regarding the measurement of emotional exhaustion, since the authors have modified the original evaluation scale (from seven points) to one of 5 points. Although this information appears much later (in table 7), it would be recommended that it appear in the description of the instruments used.
Response: There have been existing studies in the SAQ literature using a 5-point rating scale for exhaustion, and reported evidence of reliability and validity for this form of evaluation. I have now added these supporting evidence, and the reliability information from the current study when the instrument was introduced.
- In figure 1 the meaning of the initials TC, WC, PM, etc. should be added in the title or footer. The same in the tables (please, avoid using initials)
Response: These initials have now been replaced by the full terms.
- Table 7 shows mean values and standard deviations of qualitative variables !!!! Like gender, age group, managerial position, etc. This is a very serious error; with this type of variables it does not make sense to calculate these statistics. Remove these statistics
Response: These statistics have now been removed for the first 7 categorical variables in Table 7.
- The reasoning used to perform the hierarchical regression analyzes is not well understood and Table 9 is confusing. The authors should clarify how the independent variables have been introduced into the model
Response: I have now further explained the reasoning used to perform the hierarchical regression analyzes. That is, we treated the seven personal background variables as control variables (Baseline Model). Having controlled for their effects, we then entered one specific SAQ dimension in each research model (Model 1 to Model 6). For example, Model 1 shows the predictive power of teamwork climate on exhaustion after taken out the effects of personal background variables. In Model 7, the overall SAQ score was entered after con-trolling for the background variables. In Model 8, all six SAQ dimensions were entered together to compete in predicting exhaustion.

Reviewer 3 Report
I thank the Authors for the opportunity to read their article. I find the article very well written, with good theoretical knowledge and correctly conducted statistical analyzes and interpretations. I would not like to disturb the main idea presented in the article, therefore I only propose a few points for consideration.
Introduction
Do the authors consider it appropriate to add information on information security in medical facilities in the context of a safety culture? (e.g .: https://doi.org/10.1177/1460458219832048)
Perhaps readers unfamiliar with the Taiwanese medical system will benefit from adding information on how many people have received vaccinations at the moment (line 100 "Owing to Taiwanese government's swift and effective border sealing and other mass prevention measures, with the diligent cooperation of citizens across the country, Taiwan has (as to March 2021) fewer than 1,000 confirmed cases and 10 people died from COVID-19."
Materials and Methods
All key information was clearly presented. There is no need to make changes. 2.4.2. Safety Attitudes Questionnaire (SAQ) - I only miss exemplary items.
Results
The statistical calculations were correctly performed.
Line 278. Something seems to be missing (parenthesis)
Discussion and Conclusions - I have no comments.
Author Response
Revisions made following Reviewer 3’s suggestions
Introduction
- Do the authors consider it appropriate to add information on information security in medical facilities in the context of a safety culture? (e.g .: https://doi.org/10.1177/1460458219832048)
Response: At the beginning of the paper, I have now added a brief discussion on the organizational climate approaches linking information security climate and patient safety culture.
- Perhaps readers unfamiliar with the Taiwanese medical system will benefit from adding information on how many people have received vaccinations at the moment (line 100 "Owing to Taiwanese government's swift and effective border sealing and other mass prevention measures, with the diligent cooperation of citizens across the country, Taiwan has (as to March 2021) fewer than 1,000 confirmed cases and 10 people died from COVID-19."
Response: In the third paragraph from the bottom of the Introduction, I have now added COVID statistics at the time of study and that of writing up the paper.
Materials and Methods
- All key information was clearly presented. There is no need to make changes. 2.4.2. Safety Attitudes Questionnaire (SAQ) - I only miss exemplary items.
Response: I have now provided two exemplary items for the SAQ in the text, constrained by copy right. These are “I have the support I need from other personnel to care for patients” (teamwork climate) and “I am encouraged by my colleagues to report any patient safety concerns I may have” (safety climate).
- Line 278. Something seems to be missing (parenthesis)
Response: Following the statement “Overall, the differences on the endorsement of patient safety culture were most consistent and pronounced between managers and non-managerial staff”, I have now added Table 4 in (parenthesis) to direct readers to the corresponding table of results.

Round 2
Reviewer 1 Report
Thank you for addressing most of my comments.
However, I consider that the culture issue should be addressed by previous literature as indicated.
Author Response
In Line 100-111, I have now incorporated the cross-country study to highlight the culture issue of mental health during the COVID pandemic.
Murphy, M., & Moret-Tatay, C. (2021). Personality and Attitudes Confronting Death Awareness During the COVID-19 Outbreak in Italy and Spain. Frontiers in Psychiatry, 12. https://doi.org/10.3389/fpsyt.2021.627018
